# Doubly Stochastic Variational Inference
# for Deep Gaussian Processes

**Hugh Salimbeni**
Imperial College London and PROWLER.io
hrs13@ic.ac.uk

**Marc Peter Deisenroth**
Imperial College London and PROWLER.io
m.deisenroth@imperial.ac.uk

## Abstract

Gaussian processes (GPs) are a good choice for function approximation as they are flexible, robust to overfitting, and provide well-calibrated predictive uncertainty. Deep Gaussian processes (DGPs) are multi-layer generalizations of GPs, but inference in these models has proved challenging. Existing approaches to inference in DGP models assume approximate posteriors that force independence between the layers, and do not work well in practice. We present a doubly stochastic variational inference algorithm that does not force independence between layers. With our method of inference we demonstrate that a DGP model can be used effectively on data ranging in size from hundreds to a billion points. We provide strong empirical evidence that our inference scheme for DGPs works well in practice in both classification and regression.

## 1 Introduction

Gaussian processes (GPs) achieve state-of-the-art performance in a range of applications including robotics (Ko and Fox, 2008; Deisenroth and Rasmussen, 2011), geostatistics (Diggle and Ribeiro, 2007), numerics (Briol et al., 2015), active sensing (Guestrin et al., 2005) and optimization (Snoek et al., 2012). A Gaussian process is defined by its mean and covariance function. In some situations prior knowledge can be readily incorporated into these functions. Examples include periodicities in climate modelling (Rasmussen and Williams, 2006), change-points in time series data (Garnett et al., 2009) and simulator priors for robotics (Cutler and How, 2015). In other settings, GPs are used successfully as black-box function approximators. There are compelling reasons to use GPs, even when little is known about the data: a GP grows in complexity to suit the data; a GP is robust to overfitting while providing reasonable error bars on predictions; a GP can model a rich class of functions with few hyperparameters.

Single-layer GP models are limited by the expressiveness of the kernel/covariance function. To some extent kernels can be learned from data, but inference over a large and richly parameterized space of kernels is expensive, and approximate methods may be at risk of overfitting. Optimization of the marginal likelihood with respect to hyperparameters approximates Bayesian inference only if the number of hyperparameters is small (Mackay, 1999). Attempts to use, for example, a highly parameterized neural network as a kernel function (Calandra et al., 2016; Wilson et al., 2016) incur the downsides of deep learning, such as the need for application-specific architectures and regularization techniques. Kernels can be combined through sums and products (Duvenaud et al., 2013) to create more expressive compositional kernels, but this approach is limited to simple base kernels, and their optimization is expensive.

A Deep Gaussian Process (DGP) is a hierarchical composition of GPs that can overcome the limitations of standard (single-layer) GPs while retaining the advantages. DGPs are richer models than standard GPs, just as deep networks are richer than generalized linear models. In contrast to models with highly parameterized kernels, DGPs learn a representation hierarchy non-parametrically with very few hyperparmeters to optimize.

Unlike their single-layer counterparts, DGPs have proved difficult to train. The mean-field variational approaches used in previous work (Damianou and Lawrence, 2013; Mattos et al., 2016; Dai et al., 2016) make strong independence and Gaussianity assumptions. The true posterior is likely to exhibit high correlations between layers, but mean-field variational approaches are known to severely underestimate the variance in these situations (Turner and Sahani, 2011).

In this paper, we present a variational algorithm for inference in DGP models that does *not* force independence or Gaussianity between the layers. In common with many state-of-the-art GP approximation schemes we start from a sparse inducing point variational framework (Matthews et al., 2016) to achieve computational tractability *within* each layer, but we do not force independence *between* the layers. Instead, we use the exact model conditioned on the inducing points as a variational posterior. This posterior has the same structure as the full model, and in particular it maintains the correlations between layers. Since we preserve the non-linearity of the full model in our variational posterior we lose analytic tractability. We overcome this difficulty by sampling from the variational posterior, introducing the first source of stochasticity. This is computationally straightforward due to an important property of the sparse variational posterior marginals: the marginals conditioned on the layer below depend only on the corresponding inputs. It follows that samples from the marginals at the top layer can be obtained without computing the full covariance *within* the layers. We are primarily interested in large data applications, so we further subsample the data in minibatches. This second source of stochasticity allows us to scale to arbitrarily large data.

We demonstrate through extensive experiments that our approach works well in practice. We provide results on benchmark regression and classification data problems, and also demonstrate the first DGP application to a dataset with a billion points. Our experiments confirm that DGP models are never worse than single-layer GPs, and in many cases significantly better. Crucially, we show that additional layers do not incur overfitting, even with small data.

## 2 Background

In this section, we present necessary background on single-layer Gaussian processes and sparse variational inference, followed by the definition of the deep Gaussian process model. Throughout we emphasize a particular property of sparse approximations: the sparse variational posterior is itself a Gaussian process, so the marginals depend only on the corresponding inputs.

### 2.1 Single-layer Gaussian Processes

We consider the task of inferring a stochastic function $f : \mathbb{R}^D \to \mathbb{R}$, given a likelihood $p(y|f)$ and a set of $N$ observations $\mathbf{y} = (y_1, \ldots, y_N)^\top$ at design locations $\mathbf{X} = (\mathbf{x}_1, \ldots, \mathbf{x}_N)^\top$. We place a GP prior on the function $f$ that models all function values as jointly Gaussian, with a covariance function $k : \mathbb{R}^D \times \mathbb{R}^D \to \mathbb{R}$ and a mean function $m : \mathbb{R}^D \to \mathbb{R}$. We further define an additional set of $M$ inducing locations $\mathbf{Z} = (\mathbf{z}_1, \ldots, \mathbf{z}_M)^\top$. We use the notation $\mathbf{f} = f(\mathbf{X})$ and $\mathbf{u} = f(\mathbf{Z})$ for the function values at the design and inducing points, respectively. We define also $[m(\mathbf{X})]_i = m(\mathbf{x}_i)$ and $[k(\mathbf{X}, \mathbf{Z})]_{ij} = k(\mathbf{x}_i, \mathbf{z}_j)$. By the definition of a GP, the joint density $p(\mathbf{f}, \mathbf{u})$ is a Gaussian whose mean is given by the mean function evaluated at every input $(\mathbf{X}, \mathbf{Z})^\top$, and the corresponding covariance is given by the covariance function evaluated at every pair of inputs. The joint density of $\mathbf{y}, \mathbf{f}$ and $\mathbf{u}$ is

$$p(\mathbf{y}, \mathbf{f}, \mathbf{u}) = \underbrace{p(\mathbf{f}|\mathbf{u}; \mathbf{X}, \mathbf{Z})p(\mathbf{u}; \mathbf{Z})}_{\text{GP prior}} \underbrace{\prod_{i=1}^{N} p(y_i|f_i)}_{\text{likelihood}} . \tag{1}$$

In (1) we factorized the joint GP prior $p(\mathbf{f}, \mathbf{u}; \mathbf{X}, \mathbf{Z})$ [1] into the prior $p(\mathbf{u}) = \mathcal{N}(\mathbf{u}|m(\mathbf{Z}), k(\mathbf{Z}, \mathbf{Z}))$ and the conditional $p(\mathbf{f}|\mathbf{u}; \mathbf{X}, \mathbf{Z}) = \mathcal{N}(\mathbf{f}|\boldsymbol{\mu}, \boldsymbol{\Sigma})$, where for $i, j = 1, \ldots, N$

$$[\boldsymbol{\mu}]_i = m(\mathbf{x}_i) + \boldsymbol{\alpha}(\mathbf{x}_i)^\top (\mathbf{u} - m(\mathbf{Z})), \tag{2}$$

$$[\boldsymbol{\Sigma}]_{ij} = k(\mathbf{x}_i, \mathbf{x}_j) - \boldsymbol{\alpha}(\mathbf{x}_i)^\top k(\mathbf{Z}, \mathbf{Z}) \boldsymbol{\alpha}(\mathbf{x}_j), \tag{3}$$

with $\boldsymbol{\alpha}(\mathbf{x}_i) = k(\mathbf{Z}, \mathbf{Z})^{-1}k(\mathbf{Z}, \mathbf{x}_i)$. Note that the conditional mean $\boldsymbol{\mu}$ and covariance $\boldsymbol{\Sigma}$ defined via (2) and (3), respectively, take the form of mean and covariance functions of the inputs $\mathbf{x}_i$. Inference in the model (1) is possible in closed form when the likelihood $p(y|f)$ is Gaussian, but the computation scales cubically with $N$.

We are interested in large datasets with non-Gaussian likelihoods. Therefore, we seek a variational posterior to overcome both these difficulties simultaneously. Variational inference seeks an approximate posterior $q(\mathbf{f}, \mathbf{u})$ by minimizing the Kullback-Leibler divergence $\text{KL}[q||p]$ between the variational posterior $q$ and the true posterior $p$. Equivalently, we maximize the lower bound on the marginal likelihood (evidence)

$$\mathcal{L} = \mathbb{E}_{q(\mathbf{f},\mathbf{u})}\left[\log \frac{p(\mathbf{y},\mathbf{f},\mathbf{u})}{q(\mathbf{f},\mathbf{u})}\right], \tag{4}$$

where $p(\mathbf{y}, \mathbf{f}, \mathbf{u})$ is given in (1). We follow Hensman et al. (2013) and choose a variational posterior

$$q(\mathbf{f}, \mathbf{u}) = p(\mathbf{f}|\mathbf{u}; \mathbf{X}, \mathbf{Z})q(\mathbf{u}), \tag{5}$$

where $q(\mathbf{u}) = \mathcal{N}(\mathbf{u}|\mathbf{m}, \mathbf{S})$. Since both terms in the variational posterior are Gaussian, we can analytically marginalize $\mathbf{u}$, which yields

$$q(\mathbf{f}|\mathbf{m}, \mathbf{S}; \mathbf{X}, \mathbf{Z}) = \int p(\mathbf{f}|\mathbf{u}; \mathbf{X}, \mathbf{Z})q(\mathbf{u})d\mathbf{u} = \mathcal{N}(\mathbf{f}|\tilde{\boldsymbol{\mu}}, \tilde{\boldsymbol{\Sigma}}). \tag{6}$$

Similar to (2) and (3), the expressions for $\tilde{\boldsymbol{\mu}}$ and $\tilde{\boldsymbol{\Sigma}}$ can be written as mean and covariance functions of the inputs. To emphasize this point we define

$$\mu_{\mathbf{m},\mathbf{Z}}(\mathbf{x}_i) = m(\mathbf{x}_i) + \boldsymbol{\alpha}(\mathbf{x}_i)^\top (\mathbf{m} - m(\mathbf{Z})), \tag{7}$$

$$\Sigma_{\mathbf{S},\mathbf{Z}}(\mathbf{x}_i, \mathbf{x}_j) = k(\mathbf{x}_i, \mathbf{x}_j) - \boldsymbol{\alpha}(\mathbf{x}_i)^\top (k(\mathbf{Z}, \mathbf{Z}) - \mathbf{S})\boldsymbol{\alpha}(\mathbf{x}_j). \tag{8}$$

With these functions we define $[\tilde{\boldsymbol{\mu}}]_i = \mu_{\mathbf{m},\mathbf{Z}}(\mathbf{x}_i)$ and $[\tilde{\boldsymbol{\Sigma}}]_{ij} = \Sigma_{\mathbf{S},\mathbf{Z}}(\mathbf{x}_i, \mathbf{x}_j)$. We have written the mean and covariance in this way to make the following observation clear.

**Remark 1.** *The $f_i$ marginals of the variational posterior* (6) *depend only on the corresponding inputs $\mathbf{x}_i$. Therefore, we can write the $i$th marginal of $q(\mathbf{f}|\mathbf{m}, \mathbf{S}; \mathbf{X}, \mathbf{Z})$ as*

$$q(f_i|\mathbf{m}, \mathbf{S}; \mathbf{X}, \mathbf{Z}) = q(f_i|\mathbf{m}, \mathbf{S}; \mathbf{x}_i, \mathbf{Z}) = \mathcal{N}(f_i|\mu_{\mathbf{m},\mathbf{Z}}(\mathbf{x}_i), \Sigma_{\mathbf{S},\mathbf{Z}}(\mathbf{x}_i, \mathbf{x}_i)). \tag{9}$$

Using our variational posterior (5) the lower bound (4) simplifies considerably since (a) the conditionals $p(\mathbf{f}|\mathbf{u}; \mathbf{X}, \mathbf{Z})$ inside the logarithm cancel and (b) the likelihood expectation requires only the variational marginals. We obtain

$$\mathcal{L} = \sum_{i=1}^{N} \mathbb{E}_{q(f_i|\mathbf{m},\mathbf{S};\mathbf{x}_i,\mathbf{Z})}[\log p(y_i|f_i)] - \text{KL}[q(\mathbf{u})||p(\mathbf{u})]. \tag{10}$$

The final (univariate) expectation of the log-likelihood can be computed analytically in some cases, with quadrature (Hensman et al., 2015) or through Monte Carlo sampling (Bonilla et al., 2016; Gal et al., 2015). Since the bound is a sum over the data, an unbiased estimator can be obtained through minibatch subsampling. This permits inference on large datasets. In this work we refer to a GP with this method of inference as a *sparse GP (SGP)*.

The variational parameters ($\mathbf{Z}$, $\mathbf{m}$ and $\mathbf{S}$) are found by maximizing the lower bound (10). This maximization is guaranteed to converge since $\mathcal{L}$ is a lower bound to the marginal likelihood $p(\mathbf{y}|\mathbf{X})$. We can also learn model parameters (hyperparameters of the kernel or likelihood) through the maximization of this bound, though we should exercise caution as this introduces bias because the bound is not uniformly tight for all settings of hyperparameters (Turner and Sahani, 2011)

So far we have considered scalar outputs $y_i \in \mathbb{R}$. In the case of $D$-dimensional outputs $\mathbf{y}_i \in \mathbb{R}^D$ we define $\mathbf{Y}$ as the matrix with $i$th row containing the $i$th observation $\mathbf{y}_i$. Similarly, we define $\mathbf{F}$ and $\mathbf{U}$. If each output is an independent GP we have the GP prior $\prod_{d=1}^{D} p(\mathbf{F}_d|\mathbf{U}_d; \mathbf{X}, \mathbf{Z})p(\mathbf{U}_d; \mathbf{Z})$, which we abbreviate as $p(\mathbf{F}|\mathbf{U}; \mathbf{X}, \mathbf{Z})p(\mathbf{U}; \mathbf{Z})$ to lighten the notation.

## 2.2 Deep Gaussian Processes

A DGP (Damianou and Lawrence, 2013) defines a prior recursively on vector-valued stochastic functions $F^1, \ldots, F^L$. The prior on each function $F^l$ is an independent GP in each dimension, with input locations given by the noisy corruptions of the function values at the next layer: the outputs of the GPs at layer $l$ are $F_d^l$, and the corresponding inputs are $F^{l-1}$. The noise between layers is assumed i.i.d. Gaussian. Most presentations of DGPs (see, e.g. Damianou and Lawrence, 2013; Bui et al., 2016) explicitly parameterize the noisy corruptions separately from the outputs of each GP. Our method of inference does not require us to parameterize these variables separately. For notational convenience, we therefore absorb the noise into the kernel $k_{noisy}(\mathbf{x}_i, \mathbf{x}_j) = k(\mathbf{x}_i, \mathbf{x}_j) + \sigma_l^2 \delta_{ij}$, where $\delta_{ij}$ is the Kronecker delta, and $\sigma_l^2$ is the noise variance between layers. We use $D^l$ for the dimension of the outputs at layer $l$. As with the single-layer case, we have inducing locations $\mathbf{Z}^{l-1}$ at each layer and inducing function values $\mathbf{U}^l$ for each dimension.

An instantiation of the process has the joint density

$$p(\mathbf{Y}, \{\mathbf{F}^l, \mathbf{U}^l\}_{l=1}^L) = \underbrace{\prod_{i=1}^N p(\mathbf{y}_i | \mathbf{f}_i^L)}_{\text{likelihood}} \underbrace{\prod_{l=1}^L p(\mathbf{F}^l | \mathbf{U}^l; \mathbf{F}^{l-1}, \mathbf{Z}^{l-1}) p(\mathbf{U}^l; \mathbf{Z}^{l-1})}_{\text{DGP prior}}, \qquad (11)$$

where we define $\mathbf{F}^0 = \mathbf{X}$. Inference in this model is intractable, so approximations must be used.

The original DGP presentation (Damianou and Lawrence, 2013) uses a variational posterior that maintains the exact model conditioned on $\mathbf{U}^l$, but further forces the inputs to each layer to be independent from the outputs of the previous layer. The noisy corruptions are parameterized separately, and the variational distribution over these variables is a fully factorized Gaussian. This approach requires $2N(D^1 + \cdots + D^{L-1})$ variational parameters but admits a tractable lower bound on the log marginal likelihood if the kernel is of a particular form. A further problem of this bound is that the density over the outputs is simply a single layer GP with independent Gaussian inputs. Since the posterior loses all the correlations between layers it cannot express the complexity of the full model and so is likely to underestimate the variance. In practice, we found that optimizing the objective in Damianou and Lawrence (2013) results in layers being 'turned off' (the signal to noise ratio tends to zero). In contrast, our posterior retains the full conditional structure of the true model. We sacrifice analytical tractability, but due to the sparse posterior *within* each layer we can sample the bound using univariate Gaussians.

## 3 Doubly Stochastic Variational Inference

In this section, we propose a novel variational posterior and demonstrate a method to obtain unbiased samples from the resulting lower bound. The difficulty with inferring the DGP model is that there are complex correlations both *within* and *between* layers. Our approach is straightforward: we use sparse variational inference to simplify the correlations *within* layers, but we maintain the correlations *between* layers. The resulting variational lower bound cannot be evaluated analytically, but we can draw unbiased samples efficiently using univariate Gaussians. We optimize our bound stochastically.

We propose a posterior with three properties. Firstly, the posterior maintains the exact model, conditioned on $\mathbf{U}^l$. Secondly, we assume that the posterior distribution of $\{\mathbf{U}^l\}_{l=1}^L$ is factorized between layers (and dimension, but we suppress this from the notation). Therefore, our posterior takes the simple factorized form

$$q(\{\mathbf{F}^l, \mathbf{U}^l\}_{l=1}^L) = \prod_{l=1}^L p(\mathbf{F}^l | \mathbf{U}^l; \mathbf{F}^{l-1}, \mathbf{Z}^{l-1}) q(\mathbf{U}^l). \qquad (12)$$

Thirdly, and to complete specification of the posterior, we take $q(\mathbf{U}^l)$ to be a Gaussian with mean $\mathbf{m}^l$ and variance $\mathbf{S}^l$. A similar posterior was used in Hensman and Lawrence (2014) and Dai et al. (2016), but each of these works contained additional terms for the noisy corruptions at each layer.

As in the single layer SGP, we can marginalize the inducing variables from each layer analytically. After this marginalization we obtain following distribution, which is fully coupled within and between layers:

$$q(\{\mathbf{F}^l\}_{l=1}^L) = \prod_{l=1}^L q(\mathbf{F}^l | \mathbf{m}^l, \mathbf{S}^l; \mathbf{F}^{l-1}, \mathbf{Z}^{l-1}) = \prod_{l=1}^L \mathcal{N}(\mathbf{F}^l | \tilde{\boldsymbol{\mu}}^l, \tilde{\boldsymbol{\Sigma}}^l). \qquad (13)$$

Here, $q(\mathbf{F}^l|\mathbf{m}^l, \mathbf{S}^l; \mathbf{F}^{l-1}, \mathbf{Z}^{l-1})$ is as in (6). Specifically, it is a Gaussian with mean and variance $\tilde{\boldsymbol{\mu}}^l$ and $\tilde{\boldsymbol{\Sigma}}^l$, where $[\tilde{\boldsymbol{\mu}}^l]_i = \mu_{\mathbf{m}^l, \mathbf{Z}^{l-1}}(\mathbf{f}_i^l)$ and $[\tilde{\boldsymbol{\Sigma}}^l]_{ij} = \Sigma_{\mathbf{S}^l, \mathbf{Z}^{l-1}}(\mathbf{f}_i^l, \mathbf{f}_j^l)$ (recall that $\mathbf{f}_i^l$ is the $i$th row of $\mathbf{F}^l$). Since (12) is a product of terms that each take the form of the SGP variational posterior (5), we have again the property that within each layer the marginals depend on only the corresponding inputs. In particular, $\mathbf{f}_i^L$ depends only on $\mathbf{f}_i^{L-1}$, which in turn depends only on $\mathbf{f}_i^{L-2}$, and so on. Therefore, we have the following property:

**Remark 2.** *The $i$th marginal of the final layer of the variational DGP posterior* (12) *depends only on the $i$th marginals of all the other layers. That is,*

$$q(\mathbf{f}_i^L) = \int \prod_{l=1}^{L-1} q(\mathbf{f}_i^l|\mathbf{m}^l, \mathbf{S}^l; \mathbf{f}_i^{l-1}, \mathbf{Z}^{l-1}) d\mathbf{f}_i^l . \tag{14}$$

The consequence of this property is that taking a sample from $q(\mathbf{f}_i^L)$ is straightforward, and furthermore we can perform the sampling using only univariate unit Gaussians using the 're-parameterization trick' (Rezende et al., 2014; Kingma et al., 2015). Specifically, we first sample $\boldsymbol{\epsilon}_i^l \sim \mathcal{N}(\mathbf{0}, \mathbf{I}_{D^l})$ and then recursively draw the sampled variables $\hat{\mathbf{f}}_i^l \sim q(\mathbf{f}_i^l|\mathbf{m}^l, \mathbf{S}^l; \hat{\mathbf{f}}_i^{l-1}, \mathbf{Z}^{l-1})$ for $l = 1, \ldots, L-1$ as

$$\hat{\mathbf{f}}_i^l = \mu_{\mathbf{m}^l, \mathbf{Z}^{l-1}}(\hat{\mathbf{f}}_i^{l-1}) + \boldsymbol{\epsilon}_i^l \odot \sqrt{\Sigma_{\mathbf{S}^l, \mathbf{Z}^{l-1}}(\hat{\mathbf{f}}_i^{l-1}, \hat{\mathbf{f}}_i^{l-1})}, \tag{15}$$

where the terms in (15) are $D^l$-dimensional and the square root is element-wise. For the first layer we define $\hat{\mathbf{f}}_i^0 := \mathbf{x}_i$.

**Efficient computation of the evidence lower bound**    The evidence lower bound of the DGP is

$$\mathcal{L}_{DGP} = \mathbb{E}_{q(\{\mathbf{F}^l, \mathbf{U}^l\}_{l=1}^L)} \left[ \frac{p(\mathbf{Y}, \{\mathbf{F}^l, \mathbf{U}^l\}_{l=1}^L)}{q(\{\mathbf{F}^l, \mathbf{U}^l\}_{l=1}^L)} \right] . \tag{16}$$

Using (11) and (12) for the corresponding expressions in (16), we obtain after some re-arranging

$$\mathcal{L}_{DGP} = \sum_{i=1}^N \mathbb{E}_{q(\mathbf{f}_i^L)}[\log p(\mathbf{y}_n|\mathbf{f}_n^L)] - \sum_{l=1}^L \mathrm{KL}[q(\mathbf{U}^l)||p(\mathbf{U}^l; \mathbf{Z}^{l-1})], \tag{17}$$

where we exploited the exact marginalization of the inducing variables (13) and the property of the marginals of the final layer (14). A detailed derivation is provided in the supplementary material. This bound has complexity $\mathcal{O}(NM^2(D^1 + \cdots + D^L))$ to evaluate.

We evaluate the bound (17) approximately using two sources of stochasticity. Firstly, we approximate the expectation with a Monte Carlo sample from the variational posterior (14), which we compute according to (15). Since we have parameterized this sampling procedure in terms of isotropic Gaussians, we can compute unbiased gradients of the bound (17). Secondly, since the bound factorizes over the data we achieve scalability through sub-sampling the data. Both stochastic approximations are *unbiased*.

**Predictions**    To predict we sample from the variational posterior changing the input locations to the test location $\mathbf{x}_*$. We denote the function values at the test location as $\mathbf{f}_*^l$. To obtain the density over $\mathbf{f}_*^L$ we use the Gaussian mixture

$$q(\mathbf{f}_*^L) \approx \frac{1}{S} \sum_{s=1}^S q(\mathbf{f}_*^L|\mathbf{m}^L, \mathbf{S}^L; \mathbf{f}_*^{(s)L-1}, \mathbf{Z}^{L-1}), \tag{18}$$

where we draw $S$ samples $\mathbf{f}_*^{(s)L-1}$ using (15), but replacing the inputs $\mathbf{x}_i$ with the test location $\mathbf{x}_*$.

**Further Model Details**    While GPs are often used with a zero mean function, we consider such a choice inappropriate for the inner layers of a DGP. Using a zero mean function causes difficulties with the DGP prior as each GP mapping is highly non-injective. This effect was analyzed in Duvenaud et al. (2014) where the authors suggest adding the original input $\mathbf{X}$ to each layer. Instead, we consider an alternative approach and include a linear mean function $m(\mathbf{X}) = \mathbf{X}\mathbf{W}$ for all the inner layers. If the input and output dimension are the same we use the identity matrix for $\mathbf{W}$, otherwise we compute the SVD of the data and use the top $D^l$ left eigenvectors sorted by singular value (i.e. the PCA mapping). With these choices it is effective to initialize all inducing mean values $\mathbf{m}^l = \mathbf{0}$. This choice of mean function is partly inspired by the 'skip layer' approach of the ResNet (He et al., 2016) architecture.

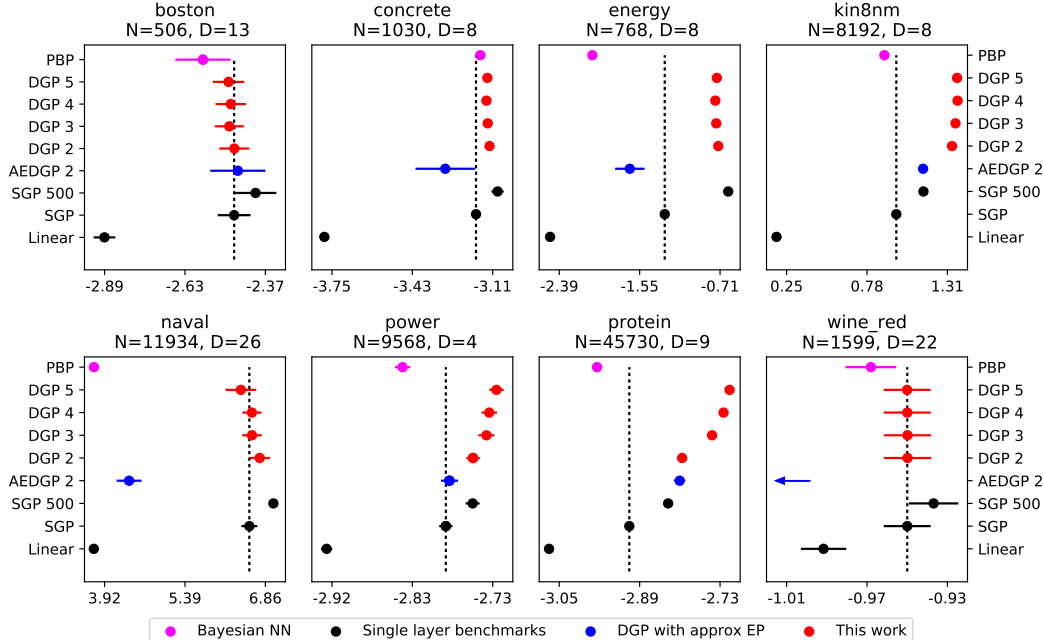

Figure 1: Regression test log-likelihood results on benchmark datasets. Higher (to the right) is better. The sparse GP with the same number of inducing points is highlighted as a baseline.

## 4 Results

We evaluate our inference method on a number of benchmark regression and classification datasets. We stress that we are interested in models that can operate in both the small and large data regimes, with little or no hand tuning. All our experiments were run with exactly the same hyperparameters and initializations. See the supplementary material for details. We use $\min(30, D^0)$ for all the inner layers of our DGP models, where $D^0$ is the input dimension, and the RBF kernel for all layers.

**Regression Benchmarks**  We compare our approach to other state-of-the-art methods on 8 standard small to medium-sized UCI benchmark datasets. Following common practice (e.g. Hernández-Lobato and Adams, 2015) we use 20-fold cross validation with a 10% randomly selected held out test set and scale the inputs and outputs to zero mean and unit standard deviation within the training set (we restore the output scaling for evaluation). While we could use any kernel, we choose the RBF kernel with a lengthscale for each dimension for direct comparison with Bui et al. (2016). The test log-likelihood results are shown in Fig. 1. We compare our models of 2, 3, 4 and 5 layers (DGP 2–5), each with 100 inducing points, with (stochastically optimized) sparse GPs (Hensman et al., 2013) with 100 and 500 inducing points points (SGP, SGP 500). We compare also to a two-layer Bayesian neural network with ReLu activations, 50 hidden units (100 for protein and year), with inference by probabilistic backpropagation (Hernández-Lobato and Adams, 2015) (PBP). The results are taken from Hernández-Lobato and Adams (2015) and were found to be the most effective of several other methods for inferring Bayesian neural networks. We compare also with a DGP model with approximate expectation propagation (EP) for inference (Bui et al., 2016). Using the authors' code [2] we ran a DGP model with 1 hidden layer using approximate expectation propagation (Bui et al., 2016) (AEPDGP 2). We used the input dimension for the hidden layer for a fair comparison with our models[3]. We found the time requirements to train a 3-layer model with this inference prohibitive. Plots for test RMSE and further results tables can be found in the supplementary material.

On five of the eight datasets, the deepest DGP model is the best. On 'wine', 'naval' and 'boston' our DGP recovers the single-layer GP, which is not surprising: 'boston' is very small, 'wine' is

near-linear (note the proximity of the linear model and the scale) and 'naval' is characterized by extremely high test likelihoods (the RMSE on this dataset is less than 0.001 for all SGP and DGP models), i.e. it is a very 'easy' dataset for a GP. The Bayesian network is not better than the sparse GP for any dataset and significantly worse for six. The Approximate EP inference for the DGP models is also not competitive with the sparse GP for many of the datasets, but this may be because the initializations were designed for lower dimensional hidden layers than we used.

Our results on these small and medium sized datasets confirm that overfitting is not observed with the DGP model, and that the DGP is never worse and often better than the single layer GP. We note in particular that on the 'power', 'protein' and 'kin8nm' datasets all the DGP models outperform the SGP with five times the number of inducing points.

**Rectangles Benchmark**    We use the Rectangle-Images dataset[4], which is specifically designed to distinguish deep and shallow architectures. The dataset consists of 12,000 training and 50,000 testing examples of size $28 \times 28$, where each image consists of a (non-square) rectangular image against a different background image. The task is to determine which of the height and width is greatest. We run 2, 3 and 4 layer DGP models, and observe increasing performance with each layer. Table 1 contains the results. Note that the 500 inducing point single-layer GP is significantly less effective than any of the deep models. Our 4-layer model achieves 77.9% classification accuracy, exceeding the best result of 77.5% reported in Larochelle et al. (2007) with a three-layer deep belief network. We also exceed the best result of 76.4% reported in Krauth et al. (2016) using a sparse GP with an Arcsine kernel, a leave-one-out objective, and 1000 inducing points.

Table 1: Results on Rectangles-Images dataset ($N = 12000$, $D = 784$)

| | Single layer GP | | Ours | | | Larochelle [2007] | | Krauth [2016] |
|---|---|---|---|---|---|---|---|---|
| | SGP | SGP 500 | DGP 2 | DGP 3 | DGP 4 | DBN-3 | SVM | SGP 1000 |
| Accuracy (%) | 76.1 | 76.4 | 77.3 | 77.8 | **77.9** | 77.5 | 76.96 | 76.4 |
| Likelihood | $-0.493$ | $-0.485$ | 0.475 | $\mathbf{-0.460}$ | $\mathbf{-0.460}$ | - | - | $-0.478$ |

**Large-Scale Regression**    To demonstrate our method on a large scale regression problem we use the UCI 'year' dataset and the 'airline' dataset, which has been commonly used by the large-scale GP community. For the 'airline' dataset we take the first 700K points for training and next 100K for testing. We use a random 10% split for the 'year' dataset. Results are shown in Table 2, with the log-likelihood reported in the supplementary material. In both datasets we see that the DGP models perform better with increased depth, significantly improving in both log likelihood and RMSE over the single-layer model, even with 500 inducing points.

Table 2: Regression test RMSE results for large datasets

| | N | D | SGP | SGP 500 | DGP 2 | DGP 3 | DGP 4 | DGP 5 |
|---|---|---|---|---|---|---|---|---|
| year | 463810 | 90 | 10.67 | 9.89 | 9.58 | 8.98 | 8.93 | **8.87** |
| airline | 700K | 8 | 25.6 | 25.1 | 24.6 | 24.3 | 24.2 | **24.1** |
| taxi | 1B | 9 | 337.5 | 330.7 | 281.4 | 270.4 | 268.0 | **266.4** |

**MNIST Multiclass Classification**    We apply the DGP with 2 and 3 layers to the MNIST multiclass classification problem. We use the robust-max multiclass likelihood (Hernández-Lobato et al., 2011) and use full unprocessed data with the standard training/test split of 60K/10K. The single-layer GP with 100 inducing points achieves a test accuracy of 97.48% and this is increased to 98.06% and 98.11% with two and three layer DGPs, respectively. The 500 inducing point single layer model achieved 97.9% in our implementation, though a slightly higher result for this model has previously been reported of 98.1% (Hensman et al., 2013) and 98.4% (Krauth et al., 2016) for the same model with 1000 inducing points. We attribute this difference to different hyperparameter initialization and training schedules, and stress that we use exactly the same initialization and learning schedule for all our models. The only other DGP result in the literature on this dataset is 94.24% (Wang et al., 2016) for a two layer model with a two dimensional latent space.

**Large-Scale Classification**   We use the HIGGS ($N = 11M$, $D = 28$) and SUSY ($N = 5.5M$, $D = 18$) datasets for large-scale binary classification. These datasets have been constructed from Monte Carlo physics simulations to detect the presence of the Higgs boson and super-symmetry (Baldi et al., 2014). We take a 10% random sample for testing and use the rest for training. We use the AUC metric for comparison with Baldi et al. (2014). Our DGP models are the highest performing on the SUSY dataset (AUC of 0.877 for all the DGP models) compared to shallow neural networks (NN, 0.875), deep neural networks (DNN, 0.876) and boosted decision trees (BDT, 0.863). On the HIGGS dataset we see a steady improvement in additional layers (0.830, 0.837, 0.841 and 0.846 for DGP 2–4 respectively). On this dataset the DGP models exceed the performance of BDT (0.810) and NN (0.816) and both single layer GP models SGP (0.785) and SGP 500 (0.794). The best performing model on this dataset is a 5 layer DNN (0.885). Full results are reported in the supplementary material.

**Massive-Scale Regression**   To demonstrate the efficacy of our model on massive data we use the New York city yellow taxi trip dataset of 1.21 billion journeys [5]. Following Peng et al. (2017) we use 9 features: time of day; day of the week; day of the month; month; pick-up latitude and longitude; drop-off latitude and longitude; travel distance. The target is to predict the journey time. We randomly select 1B ($10^9$) examples for training and use 1M examples for testing, and we scale both inputs and outputs to zero mean and unit standard deviation in the training data. We discard journeys that are less than $10\,\mathrm{s}$ or greater than $5\,\mathrm{h}$, or start/end outside the New York region, which we estimate to have squared distance less than $5^o$ from the center of New York. The test RMSE results are the bottom row of Table 2 and

Table 3: Typical computation time in seconds for a single gradient step.

|         | CPU  | GPU   |
|---------|------|-------|
| SGP     | 0.14 | 0.018 |
| SGP 500 | 1.71 | 0.11  |
| DGP 2   | 0.36 | 0.030 |
| DGP 3   | 0.49 | 0.045 |
| DGP 4   | 0.65 | 0.056 |
| DGP 5   | 0.87 | 0.069 |

test log likelihoods are in the supplementary material. We note the significant jump in performance from the single layer models to the DGP. As with all the large-scale experiments, we see a consistent improvement extra layers, but on this dataset the improvement is particularly striking (DGP 5 achieves a 21% reduction in RMSE compared to SGP)

## 5   Related Work

The first example of the outputs of a GP used as the inputs to another GP can be found in Lawrence and Moore (2007). MAP approximation was used for inference. The seminal work of Titsias and Lawrence (2010) demonstrated how sparse variational inference could be used to propagate Gaussian inputs through a GP with a Gaussian likelihood. This approach was extended in Damianou et al. (2011) to perform approximate inference in the model of Lawrence and Moore (2007), and shortly afterwards in a similar model Lázaro-Gredilla (2012), which also included a linear mean function. The key idea of both these approaches is the factorization of the variational posterior *between* layers. A more general model (flexible in depth and dimensions of hidden layers) introduced the term 'DGP' and used a posterior that also factorized *between* layers. These approaches require a linearly increasing number of variational parameters in the number of data. For high-dimensional observations, it is possible to amortize the cost of this optimization with an auxiliary model. This approach is pursued in Dai et al. (2016), and with a recurrent architecture in Mattos et al. (2016). Another approach to inference in the exact model was presented in Hensman and Lawrence (2014), where a sparse approximation was used within layers for the GP outputs, similar to Damianou and Lawrence (2013), but with a projected distribution over the inputs to the next layer. The particular form of the variational distribution was chosen to admit a tractable bound, but imposes a constraint on the flexibility.

An alternative approach is to modify the DGP prior directly and perform inference in a parametric model. This is achieved in Bui et al. (2016) with an inducing point approximation within each layer, and in Cutajar et al. (2017) with an approximation to the spectral density of the kernel. Both approaches then apply additional approximations to achieve tractable inference. In Bui et al. (2016), an approximation to expectation propagation is used, with additional Gaussian approximations to the log partition function to propagate uncertainly through the non-linear GP mapping. In Cutajar et al. (2017) a fully factorized  variational approximation is used for the spectral components. Both these

approaches require specific kernels: in Bui et al. (2016) the kernel must have analytic expectations under a Gaussian, and in Cutajar et al. (2017) the kernel must have an analytic spectral density. Vafa (2016) also uses the same initial approximation as Bui et al. (2016) but applies MAP inference for the inducing points, such that the uncertainty propagated through the layers only represents the quality of the approximation. In the limit of infinitely many inducing points this approach recovers a deterministic radial basis function network. A particle method is used in Wang et al. (2016), again employing an online version of the sparse approximation used by Bui et al. (2016) within each layer. Similarly to our approach, in Wang et al. (2016) samples are taken through the conditional model, but differently from us they then use a point estimate for the latent variables. It is not clear how this approach propagates uncertainty through the layers, since the GPs at each layer have point-estimate inputs and outputs.

A pathology with the DGP with zero mean function for the inner layers was identified in Duvenaud et al. (2014). In Duvenaud et al. (2014) a suggestion was made to concatenate the original inputs at each layer. This approach is followed in Dai et al. (2016) and Cutajar et al. (2017). The linear mean function was original used by Lázaro-Gredilla (2012), though in the special case of a two layer DGP with a 1D hidden layer. To the best of our knowledge there has been no previous attempt to use a linear mean function for all inner layers.

## 6 Discussion

Our experiments show that on a wide range of tasks the DGP model with our doubly stochastic inference is both effective and scalable. Crucially, we observe that on the small datasets the DGP does not overfit, while on the large datasets additional layers generally increase performance and never deteriorate it. In particular, we note that the largest gain with increasing layers is achieved on the largest dataset (the taxi dataset, with 1B points). We note also that on all the large scale experiments the SGP 500 model is outperformed by the *all* the DGP models. Therefore, for the same computational budget increasing the number of layers can be significantly more effective than increasing the accuracy of approximate inference in the single-layer model. Other than the additional computation time, which is fairly modest (see Table 3), we do not see downsides to using a DGP over a single-layer GP, but substantial advantages.

While we have considered simple kernels and black-box applications, any domain-specific kernel could be used in any layer. This is in contrast to other methods (Damianou and Lawrence, 2013; Bui et al., 2016; Cutajar et al., 2017) that require specific kernels and intricate implementations. Our implementation is simple ($< 200$ lines), publicly available [6], and is integrated with GPflow (Matthews et al., 2017), an open-source GP framework built on top of Tensorflow (Abadi et al., 2015).

## 7 Conclusion

We have presented a new method for inference in Deep Gaussian Process (DGP) models. With our inference we have shown that the DGP can be used on a range of regression and classification tasks with no hand-tuning. Our results show that in practice the DGP always exceeds or matches the performance of a single layer GP. Further, we have shown that the DGP often exceeds the single layer significantly, even when the quality of the approximation to the single layer is improved. Our approach is highly scalable and benefits from GPU acceleration.

The most significant limitation of our approach is the dealing with high dimensional inner layers. We used a linear mean function for the high dimensional datasets but left this mean function fixed, as to optimize the parameters would go against our non-parametric paradigm. It would be possible to treat this mapping probabilistically, following the work of Titsias and Lázaro-Gredilla (2013).

**Acknowledgments**

We have greatly appreciated valuable discussions with James Hensman and Steindor Saemundsson in the preparation of this work. We thank Vincent Dutordoir and anonymous reviewers for helpful feedback on the manuscript. We are grateful for a Microsoft Azure Scholarship and support through a Google Faculty Research Award to Marc Deisenroth.

## Footnotes

[1] Throughout this paper we use the semi-colon notation to clarify the input locations of the corresponding function values, which will become important later when we discuss multi-layer GP models. For example, $p(\mathbf{f}|\mathbf{u}; \mathbf{X}, \mathbf{Z})$ indicates that the input locations for $\mathbf{f}$ and $\mathbf{u}$ are $\mathbf{X}$ and $\mathbf{Z}$, respectively.

[2] `https://github.com/thangbui/deepGP_approxEP`

[3] We note however that in Bui et al. (2016) the inner layers were 2D, so the results we obtained are not directly comparable to those reported in Bui et al. (2016)

[4]`http://www.iro.umontreal.ca/~lisa/twiki/bin/view.cgi/Public/RectanglesData`

[5]`http://www.nyc.gov/html/tlc/html/about/trip_record_data.shtml`

[6]`https://github.com/ICL-SML/Doubly-Stochastic-DGP`

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
