[Supplementary Material]

## Supplementary Material

## Experiment Details

**Variational parameters initializations** All our DGP models have 100 inducing points, initialized with K-means (computed from a random 100M subset for the taxi data). The inducing function values means are all initialized to zero, and variances to the identity, which we scale by $10^{-5}$ for the inner layers.

**Model hyperparameter initializations** We initialize all kernel variances and lengthscales to 2 at every layer, and initialize the likelihood variance to 0.01. We initialize the noise between the layers (separately for each layer) to $10^{-5}$.

**Training** We optimize all hyperparameters and variational parameters jointly. We use a minibatch size of 10K (or the size of the data for the datasets with fewer than 10K points). We use the Adam optimizer (Kingma and Ba, 2015) with a learning rate of 0.01 with all other parameters set to the default values. We train for 20,000 iterations for the small to medium regression datasets, and 100,000 for the others (500,000 for the taxi dataset, which is 5 epochs)

**Computation** Our implementation is based on GPflow Matthews et al. (2017) and leverages automatic differentiation and GPU acceleration in Tensorflow (Abadi et al., 2015). We use Azure NC6 instances with Tesla K80 GPUs for all computations. The GPU implementation speeds up computation by an order of magnitude. See Table 4 (repeated here from the main text) for timing results for CPU (8 core i5) and GPU (Tesla K80). These timing results are for a minibatch size of 10000, with inner dimensions all equal to one, averaged over 100 steps. Note that we achieve slightly sub-linear scaling in depth.

Table 4: Typical computation time in seconds for a single gradient step

|  | CPU | GPU |
|---|---|---|
| SGP | 0.14 | 0.018 |
| SGP 500 | 1.71 | 0.11 |
| DGP 2 | 0.36 | 0.030 |
| DGP 3 | 0.49 | 0.045 |
| DGP 4 | 0.65 | 0.056 |
| DGP 5 | 0.87 | 0.069 |

## Further results

Figure 2: Regression test RMSE results on benchmark datasets. Lower (to the left) is better. The mean is shown with error bars of one standard error. The sparse GP with the same number of inducing points is highlighted as a baseline.

Table 5: Regression test log likelihood results for large datasets

|         | N      | D  | SGP   | SGP 500 | DGP 2 | DGP 3 | DGP 4 | DGP 5     |
|---------|--------|----|-------|---------|-------|-------|-------|-----------|
| year    | 463810 | 90 | −3.74 | −3.65   | −3.63 | −3.57 | −3.56 | **−3.32** |
| airline | 700K   | 8  | −4.66 | −4.63   | −4.61 | −4.59 | −4.59 | **−4.58** |
| taxi    | 1B     | 9  | −7.24 | −7.22   | −7.06 | −7.02 | −7.01 | **−7.00** |

Table 6: Binary classification AUC results on high energy physics data. We report vales for comparison with Baldi et al. (2014)

|       |      |    | Single layer GP | | Ours | | | | Other reported results | | |
|-------|------|----|------|---------|-------|-------|-------|-------|-------|-------|-----------|
|       | N    | D  | SGP   | SGP 500 | DGP 2 | DGP 3 | DGP 4 | DGP 5 | BDT   | NN    | DNN       |
| HIGGS | 11M  | 24 | 0.785 | 0.794   | 0.830 | 0.837 | 0.841 | 0.846 | 0.810 | 0.816 | **0.885** |
| SUSY  | 5.5M | 18 | 0.875 | 0.876   | **0.877** | **0.877** | **0.877** | **0.877** | 0.863 | 0.875 | 0.876 |

Table 7: Regression test log likelihood results. Reported is the mean over 20 splits (with standard errors)

| | N | D | Linear | SGP | SGP 500 | AEPDGP 2 | DGP 2 | DGP 3 | DGP 4 | DGP 5 | PBP |
|---|---|---|---|---|---|---|---|---|---|---|---|
| boston | 506 | 13 | −2.89(0.03) | **−2.47(0.05)** | **−2.40(0.07)** | **−2.46(0.09)** | **−2.47(0.05)** | −2.49(0.05) | −2.48(0.05) | −2.49(0.05) | **−2.57(0.09)** |
| concrete | 1030 | 8 | −3.78(0.01) | −3.18(0.02) | **−3.09(0.02)** | −3.30(0.12) | −3.12(0.01) | −3.13(0.01) | −3.14(0.01) | −3.13(0.01) | −3.16(0.02) |
| energy | 768 | 8 | −2.48(0.02) | −1.29(0.02) | **−0.63(0.03)** | −1.65(0.15) | −0.73(0.02) | −0.75(0.02) | −0.76(0.02) | −0.74(0.02) | −2.04(0.02) |
| kin8nm | 8192 | 8 | 0.18(0.01) | 0.97(0.00) | 1.15(0.00) | 1.15(0.03) | 1.34(0.01) | 1.37(0.01) | **1.38(0.01)** | **1.38(0.01)** | 0.90(0.01) |
| naval | 11934 | 26 | 3.73(0.00) | 6.57(0.15) | **7.01(0.05)** | 4.37(0.23) | 6.76(0.19) | 6.62(0.18) | 6.61(0.17) | 6.41(0.28) | 3.73(0.01) |
| power | 9568 | 4 | −2.93(0.01) | −2.79(0.01) | −2.75(0.01) | −2.78(0.01) | −2.75(0.01) | **−2.74(0.01)** | **−2.74(0.01)** | **−2.73(0.01)** | −2.84(0.01) |
| protein | 45730 | 9 | −3.07(0.00) | −2.91(0.00) | −2.83(0.00) | −2.81(0.01) | −2.81(0.00) | −2.75(0.00) | −2.73(0.00) | **−2.71(0.00)** | −2.97(0.00) |
| wine_red | 1599 | 22 | −0.99(0.01) | −0.95(0.01) | **−0.93(0.01)** | −1.51(0.09) | −0.95(0.01) | −0.95(0.01) | −0.95(0.01) | −0.95(0.01) | −0.97(0.01) |

Table 8: Regression test RMSE results

| | N | D | Linear | SGP | SGP 500 | AEPDGP 2 | DGP 2 | DGP 3 | DGP 4 | DGP 5 | PBP |
|---|---|---|---|---|---|---|---|---|---|---|---|
| boston | 506 | 13 | 4.24(0.16) | 2.87(0.15) | **2.73(0.12)** | 3.42(0.37) | 2.90(0.17) | 2.93(0.16) | 2.90(0.15) | 2.92(0.17) | 3.01(0.18) |
| concrete | 1030 | 8 | 10.54(0.13) | 5.97(0.11) | **5.53(0.12)** | 7.68(0.90) | **5.61(0.10)** | **5.64(0.10)** | 5.68(0.10) | 5.65(0.10) | 5.67(0.09) |
| energy | 768 | 8 | 2.88(0.05) | 0.78(0.02) | **0.47(0.02)** | 1.70(0.42) | **0.47(0.01)** | **0.48(0.01)** | **0.48(0.01)** | **0.47(0.01)** | 1.80(0.05) |
| kin8nm | 8192 | 8 | 0.20(0.00) | 0.09(0.00) | 0.08(0.00) | 0.08(0.00) | **0.06(0.00)** | **0.06(0.00)** | **0.06(0.00)** | **0.06(0.00)** | 0.10(0.00) |
| naval | 11934 | 26 | 0.01(0.00) | **0.00(0.00)** | **0.00(0.00)** | **0.00(0.00)** | **0.00(0.00)** | **0.00(0.00)** | **0.00(0.00)** | **0.00(0.00)** | 0.01(0.00) |
| power | 9568 | 4 | 4.51(0.03) | 3.91(0.03) | 3.79(0.03) | 3.99(0.03) | 3.79(0.03) | 3.73(0.04) | **3.71(0.04)** | **3.68(0.03)** | 4.12(0.03) |
| protein | 45730 | 9 | 5.21(0.02) | 4.43(0.03) | 4.10(0.03) | 4.54(0.02) | 4.00(0.03) | 3.81(0.04) | 3.74(0.04) | **3.72(0.04)** | 4.73(0.01) |
| wine_red | 1599 | 22 | 0.65(0.01) | 0.63(0.01) | **0.62(0.01)** | 0.64(0.01) | 0.63(0.01) | 0.63(0.01) | 0.63(0.01) | 0.63(0.01) | 0.64(0.01) |

## Derivation of the Lower Bound

The evidence lower bound of our DGP model is given by

$$\mathcal{L}_{DGP} = \mathbb{E}_{q(\{\mathbf{F}^l, \mathbf{U}^l\}_{l=1}^L)}\left[\frac{p(\mathbf{Y}, \{\mathbf{F}^l, \mathbf{U}^l\}_{l=1}^L)}{q(\{\mathbf{F}^l, \mathbf{U}^l\}_{l=1}^L)}\right],$$

$$q(\{\mathbf{F}^l, \mathbf{U}^l\}_{l=1}^L) = \prod_{l=1}^L p(\mathbf{F}^l|\mathbf{U}^l; \mathbf{F}^{l-1}; \mathbf{Z}^{l-1})q(\mathbf{U}^l)$$

$$p(\mathbf{Y}, \{\mathbf{F}^l, \mathbf{U}^l\}_{l=1}^L) = \underbrace{\prod_{i=1}^N p(\mathbf{y}_i|\mathbf{f}_i^L)}_{\text{likelihood}}\underbrace{\prod_{l=1}^L p(\mathbf{F}^l|\mathbf{U}^l; \mathbf{F}^{l-1}, \mathbf{Z}^{l-1})p(\mathbf{U}^l; \mathbf{Z}^{l-1})}_{\text{DGP prior}}$$

Therefore,

$$\mathcal{L}_{DGP} = \iint q(\{\mathbf{F}^l, \mathbf{U}^l\}_{l=1}^L)\log\left(\frac{p(\mathbf{Y}, \{\mathbf{F}^l, \mathbf{U}^l\}_{l=1}^L)}{q(\{\mathbf{F}^l, \mathbf{U}^l\}_{l=1}^L)}\right)d\{\mathbf{F}^l, \mathbf{U}^l\}_{l=1}^L$$

$$= \iint q(\{\mathbf{F}^l, \mathbf{U}^l\}_{l=1}^L)$$

$$\log\left(\frac{\prod_{i=1}^N p(\mathbf{y}_i|\mathbf{f}_i^L)\prod_{l=1}^L p(\mathbf{F}^l|\mathbf{U}^l; \mathbf{F}^{l-1}, \mathbf{Z}^{l-1})p(\mathbf{U}^l; \mathbf{Z}^{l-1})}{\prod_{l=1}^L p(\mathbf{F}^l|\mathbf{U}^l; \mathbf{F}^{l-1}; \mathbf{Z}^{l-1})q(\mathbf{U}^l)}\right)d\{\mathbf{F}^l, \mathbf{U}^l\}_{l=1}^L$$

We see that terms inside the logarithm cancel out, such that we obtain

$$\mathcal{L}_{DGP} = \iint q(\{\mathbf{F}^l, \mathbf{U}^l\}_{l=1}^L)\log\left(\frac{\prod_{i=1}^N p(\mathbf{y}_i|\mathbf{f}_i^L)\prod_{l=1}^L p(\mathbf{U}^l; \mathbf{Z}^{l-1})}{\prod_{l=1}^L q(\mathbf{U}^l)}\right)d\{\mathbf{F}^l, \mathbf{U}^l\}_{l=1}^L$$

$$= \iint q(\{\mathbf{F}^l, \mathbf{U}^l\}_{l=1}^L)\log\left(\prod_{i=1}^N p(\mathbf{y}_i|\mathbf{f}_i^L)\right)d\{\mathbf{F}^l, \mathbf{U}^l\}_{l=1}^L$$

$$+ \iint q(\{\mathbf{F}^l, \mathbf{U}^l\}_{l=1}^L)\log\left(\frac{\prod_{l=1}^L p(\mathbf{U}^l; \mathbf{Z}^{l-1})}{\prod_{l=1}^L q(\mathbf{U}^l)}\right)d\{\mathbf{F}^l, \mathbf{U}^l\}_{l=1}^L$$

$$= \int q(\{\mathbf{F}^l\}_{l=1}^L)\log\left(\prod_{i=1}^N p(\mathbf{y}_i|\mathbf{f}_i^L)\right)d\{\mathbf{F}^l\}_{l=1}^L$$

$$+ \int q(\{\mathbf{U}^l\}_{l=1}^L)\log\left(\frac{\prod_{l=1}^L p(\mathbf{U}^l; \mathbf{Z}^{l-1})}{\prod_{l=1}^L q(\mathbf{U}^l)}\right)d\{\mathbf{U}^l\}_{l=1}^L$$

$$= \mathbb{E}_{q(\mathbf{f}_i^L)}[\log p(y_i|\mathbf{f}_i^L)] - \sum_{l=1}^L KL\big(q(\mathbf{U}^l)||p(\mathbf{U}^l; \mathbf{Z}^{l-1})\big)$$