[Reviews · NeurIPS 2017]

Reviewer 1



This paper presents a doubly stochastic variational inference for deep Gaussian processes. The first source of stochasticity comes from posterior sampling, devised to solve the intractability issue of propagating uncertainty through the layers. The second source of stochasticity comes from minibatch learning, allowing the method to scale to big datasets, even with 1 billion instances. The paper is in general well written and the authors make an effort to explain the intuitions behind the proposed method. However, the paper is not placed well in the related literature. The "related work" section is missing reference to [11] (presumably it was meant to be included in line 65) and the paper is completely missing reference to the very related (Hensman and Lawrence, 2014; Nested Variational Compression in DGPs). The idea of using SVI and mini-batch learning for DGPs has also been shown in (Frigola et al. 2014; Variational GP State-Space Models) and (Damianou, 2015; PhD Thesis). To further expand on the above, it would be easier for the reader if the authors explained modeling and inference choices with respect to other DGP approaches. For example, using the exact conditional p(f|u) to obtain the cancellation is found in a number of approaches, but in the paper this is somehow presented as a new element. On the other hand, absorbing the noise \epsilon in the kernel is (to my knowledge) unique to this approach, however this design choice is underplayed (same holds for the mean function), stating that "it is for notational convenience", although in practice it has implications that go beyond that (e.g. no need to have q(h), h being the intermediate noisy layer). This is counter-intuitive, because these are actually neat ideas and I'd expect them to be surfaced more. Therefore, I suggest to the authors to clearly relate these details to other DGP approaches, to facilitate comprehension and theoretical comparison. Regarding the technical aspect, I find the method sound and significant. The method is not very novel, especially given (Hensman and Lawrence, 2014), but it does introduce some nice new ideas and shows extensive and impressive results. It is perhaps the first paper promising "out-of-the-box" training of DGPs and this is a significant step forward - although given this strong claim it'd be great to also have the code at submission time, especially since it's < 200 lines. It would be worth discussing or exploring some further properties of the method. For example, what does it imply to absorb the layer noise in the kernel? How does the model construction and sampling for inference affect the uncertainty calibration? I find the experiments extensive, well executed and convincing. However, I didn't understand what line 223 means ("we used the input dimension..") and why can't we see a deeper AEPDGP (does that choice make it fair for one model but unfair for the other?). Since the method is relatively fast (Table 3) and performance improves with the number of layers, this begs the question of "how deep can you go?". It'd be great to see a deeper architecture in the final version (in [8] the authors make a similar observation about the number of layers). Other comments: - including an algorithm showing the steps of the sampling-based inference would greatly improve clarity - line 176: Shouldn't f_i^l be f_i^{l-1} before "(recall..."? - line 211: It's not clear to me what "whiten inputs and outputs" means and why this is done - line 228: What does "our DGP falls back to the single-layer GP" mean? Learning the identity mapping between layers? - line 264: Reference missing. - line 311: Does "even when the quality of the approximation..." refer to having more inducing points per layer, or is it a hint about the within-the-layer approximation? - which kernel is used for classification?

Reviewer 2



The paper addresses the problem of approximate inference in deep GP models. The authors propose to use a Gaussian variational approximation to the posterior distribution over the latent function values at some inducing inputs. This approximation is then adjusted by minimizing the variational loss. The key difference of the proposed work with respect to other previous approaches is that the authors perform sampling to approximation the expected log-likelihood by Monte Carlo. For each data point in a minibatch, the authors sample the corresponding values of the latent functions at the first layer evaluated at that input and then repeat the process in subsequent layers. The advantage of this is that they avoid Gaussian approximations used by previous approaches and they avoid having to assume independence between the different values of the latent functions at different layers for a particular input. The experiments performed illustrate the superior performance of the proposed approach with respect to existing techniques. Quality The paper seems to be sound. My only critique is that the authors use models with a very large number of latent functions at each layer of the deep GP network. They use min(30, D) where D is the dimensionality of the original data. In the problems from figure 1 this means that typical values for the number of latent functions are 13, 8, 8, 8, 26, 4, 9 and 22. These are many more functions that necessary and can produce significant overfitting problems in most cases. This can be noticed when one compares the numbers for test log-likelihood reported in Figure 1 with those reported by Bui et al. 2016, which are higher in almost all the analyzed data sets. I would suggest the authors to repeat the experiments using a smaller number of latent functions in the hidden layers of the deep GP network. For example, Bui et al. 2016 use 2 or 3 functions in the hidden layers. Why did the authors not use a smaller number of hidden functions? Clarity The paper is clearly written and easy to read. Originality The proposed method is original and different from previous work. It is also expected to work better in practice by eliminating the biases of previous approaches. Significance The proposed method seems to be the best existing technique for approximate inference in deep Gaussian process models and is therefore very significant. Minor comments: - clarify what dimension is in line 169. - y_n and f_n^L should have subscript i instead of n in equation 16.

Reviewer 3



The paper proposes a new variational approximation for deep Gaussian processes (DGPs) that captures some dependence between the latent GP functions across layers. This is based on augmenting each layer-specific GP with inducing variables (as done in the standard sparse variational approximation) and re-using each GP conditional prior (that has as inputs the function values from the previous layer) in the variational distribution. This leads to a quite simple lower bound that involves intractable expectations over the log likelihood terms that can be dealt with during optimization based on the reparametrization trick. The paper is clearly written. The variational approximation used in this paper is better than the one from the original DGP paper and at the same time is somehow simpler, in the sense that it leads to a simpler expression for the lower bound. A nice feature of the approach is that the layer-specific noisy corruptions are not necessary for the approximation to work and also arbitrary kernel functions can be easily accommodated. The experimental comparison is quite comprehensive and it shows that the method is promising. Something I didn't see in the experiments is a comparison with the original variational approximation of Damianou and Lawrence. Where are the results of this earlier approximation?